# LEARNING GOAL-CONDITIONED VALUE FUNCTIONS WITH ONE-STEP PATH REWARDS RATHER THAN GOAL-REWARDS

## ABSTRACT

Multi-goal reinforcement learning (MGRL) addresses tasks where the desired goal state can change for every trial. State-of-the-art algorithms model these problems such that the reward formulation depends on the goals, to associate them with high reward. This dependence introduces additional goal reward resampling steps in algorithms like Hindsight Experience Replay (HER) that reuse trials in which the agent fails to reach the goal by recomputing rewards as if reached states were psuedo-desired goals. We propose a reformulation of goal-conditioned value functions for MGRL that yields a similar algorithm, while removing the dependence of reward functions on the goal. Our formulation thus obviates the requirement of reward-recomputation that is needed by HER and its extensions. We also extend a closely related algorithm, Floyd-Warshall Reinforcement Learning, from tabular domains to deep neural networks for use as a baseline. Our results are competitive with HER while substantially improving sampling efficiency in terms of reward computation.

## 1 INTRODUCTION

Many tasks in robotics require the specification of a *goal* for every trial. For example, a robotic arm can be tasked to move an object to an arbitrary goal position on a table (Gu et al., 2017); a mobile robot can be tasked to navigate to an arbitrary goal landmark on a map (Zhu et al., 2017). The adaptation of reinforcement learning to such goal-conditioned tasks where goal locations can change is called Multi-Goal Reinforcement Learning (MGRL) (Plappert et al., 2018). State-of-the-art MGRL algorithms (Andrychowicz et al., 2017; Pong et al., 2018) work by estimating *goal-conditioned value functions* (GCVF) which are defined as expected cumulative rewards from start states with specified goals. GCVFs, in turn, are used to compute *policies* that determine the actions to take at every state.

To learn GCVFs, MGRL algorithms use *goal-reward*, defined as the relatively higher reward recieved on reaching the desired goal state. This makes the reward function dependent on the desired goal. For example, in the Fetch-Push task (Plappert et al., 2018) of moving a block to a given location on a table, every movement incurs a "-1" reward while reaching the desired goal returns a "0" goal-reward. This dependence introduces additional reward resampling steps in algorithms like Hindsight Experience Replay (HER) (Andrychowicz et al., 2017), where trials in which the agent failed to reach the goal are reused by recomputing rewards as if the reached states were pseudo-desired goals. Due to the dependence of the reward function on the goal, the relabelling of every pseudo-goal requires an independent *reward-recomputation* step, which can be expensive.

In this paper, we demonstrate that *goal-rewards are not needed to learn GCVFs*. For the Fetch-Push example, the "0" goal-reward does not need to be achieved to learn its GCVF. Specifically, the agent continues to receive "-1" reward even when the block is in the given goal location. This reward formulation is atypical in conventional RL because high reward is used to specify the desired goal location. However, this goal-reward is not necessary in goal-conditioned RL because the goal is already specified at the start of every episode. We use this idea, to propose a goal-conditioned RL algorithm which learns to reach goals without goal-rewards. This is a counter-intuitive result which is important for understanding goal-conditioned RL.

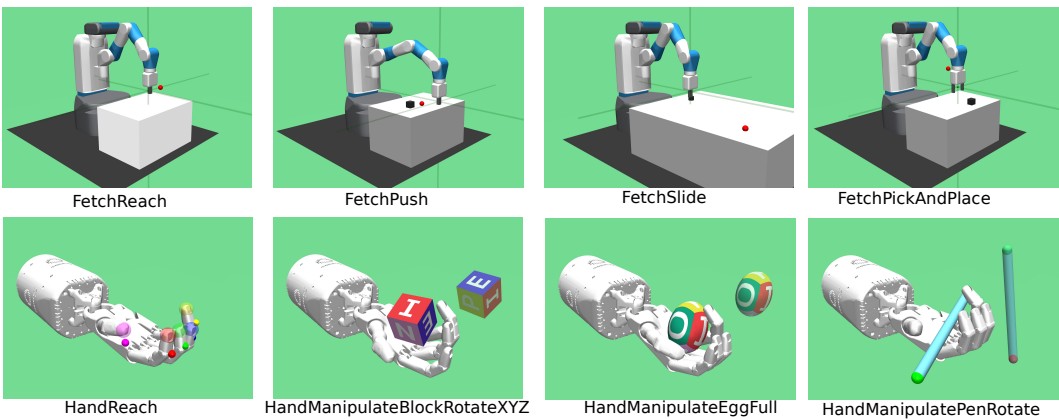

Figure 1: Plappert et al. (2018) introduce challenging tasks on the Fetch robot and the Shadow Dextrous hand. We use these tasks for our experiments. Images are taken from the technical report.

Let us consider another example to motivate the redundancy of goal-rewards. Consider a student who has moved to a new campus. To learn about the campus, the student explores it randomly with no specific goal in mind. The key intuition here is that the student is not incentivized to find specific goal locations (i.e. no goal-rewards) but is aware of the effort required to travel between points around the university. When tasked with finding a goal classroom, the student can chain together these path efforts to find the least-effort path to the classroom. Based on this intuition of least-effort paths, we redefine GCVFs to be the expected *path-reward* that is learned for all possible start-goal pairs. We introduce a *one-step loss* that assumes one-step paths to be the paths of maximum reward between pairs wherein the state and goal are adjacent. Under this interpretation, the *Bellman equation* chooses and chains together one-step paths to find longer maximum reward paths. Experimentally, we show how this simple reinterpretation, which does not use goal rewards, performs as well as HER while outperforming it in terms of reward computation.

We also extend a closely related algorithm, Floyd-Warshall Reinforcement Learning (FWRL) (Kaelbling, 1993) (also called Dynamic Goal Reinforcement learning) to use parametric function approximators instead of tabular functions. Similar to our re-definition of GCVFs, FWRL learns a goal-conditioned Floyd-Warshall function that represents path-rewards instead of future-rewards. We translate FWRL's compositionality constraints in the space of GCVFs to introduce additional loss terms to the objective. However, these additional loss terms do not show improvement over the baseline. We conjecture that the compositionality constraints are already captured by other loss terms.

In summary, the contributions of this work are twofold. Firstly, we reinterpret goal-conditioned value functions as expected path-rewards and introduce one-step loss, thereby removing the dependency of GCVFs on goal-rewards and reward resampling. We showcase our algorithm's improved sample efficiency (in terms of reward computation). We thus extend algorithms like HER to domains where reward recomputation is expensive or infeasible. Secondly, we extend the tabular Floyd-Warshal Reinforcement Learning to use deep neural networks.

## 2 RELATED WORK

Goal-conditioned tasks in reinforcement learning have been approached in two ways, depending upon whether the algorithm explicitly separates state and goal representations. The first approach is to use vanilla reinforcement learning algorithms that do not explicitly make this separation (Mirowski et al., 2016; Dosovitskiy & Koltun, 2016; Gupta et al., 2017; Parisotto & Salakhutdinov, 2017; Mirowski et al., 2018). These algorithms depend upon neural network architectures to carry the burden of learning the separated representations.

The second approach makes this separation explicit via the use of goal-conditioned value functions (Foster & Dayan, 2002; Sutton et al., 2011). *Universal Value Function Appoximators* (Schaul et al., 2015) propose a network architecture and a factorization technique that separately encodes states

and goals, taking advantage of correlations in their representations. *Temporal Difference Models* combine model-free and model-based RL to gain advantages from both realms by defining and learning a horizon-dependent GCVF. All these works require the use of goal-dependent reward functions and define GCVFs as future-rewards instead of path-rewards, contrasting them from our contribution.

Unlike our approach, Andrychowicz et al. (2017) propose *Hindsight Experience Replay*, a technique for resampling state-goal pairs from failed experiences; which leads to faster learning in the presence of sparse rewards. In addition to depending on goal rewards, HER also requires the repeated recomputation of the reward function. In contrast, we show how removing goal-rewards removes the need for such recomputations. We utilize HER as a baseline in our work.

Kaelbling (1993) also use the structure of the space of GCVFs to learn. This work employs compositionality constraints in the space of these functions to accelerate learning in a tabular domain. While their definition of GCVFs is similar to ours, the terminal condition is different. We describe this difference in Section 4. We also extend their tabular formulation to deep neural networks and evaluate it against the baselines.

## 3 BACKGROUND

A reinforcement learning (RL) problem is formalized as a Markov Decision Process (MDP) (Sutton et al., 1998). A MDP is defined by a five tuple $(\mathcal{S}, \mathcal{A}, T, R, \gamma)$, that governs a sequence of state-action-reward triples $[(s_0, a_0, r_0), \ldots, (s_T, a_T, r_T)]$. $\mathcal{S}$ is the state space, $\mathcal{A}$ is the action space, $T(s, a) : \mathcal{S} \times \mathcal{A} \to \mathcal{S}$ is the system dynamics, $R(s, a) : \mathcal{S} \times \mathcal{A} \to \mathbb{R}$ is the reward function and $\gamma$ is the discount factor. In a typical RL problem the transition function $T$ is not given but is known to be static. In RL, the objective is to find a policy $\pi(s) : \mathcal{S} \to \mathcal{A}$ that maximizes the expected cumulative reward over time, $R_t = \sum_{k=t}^{\infty} \gamma^{k-t} r_k$, called the *return*. The discount factor, $\gamma < 1$, forces the return to be finite for infinite horizons. Reinforcement learning is typically formulated in single-goal contexts. More recently there has been interest in multi-goal problems (Andrychowicz et al., 2017; Pong et al., 2018; Plappert et al., 2018), which is the focus of this work.

### 3.1 DEEP REINFORCEMENT LEARNING

A number of reinforcement learning algorithms use parametric function approximators to estimate the return in the form of an action-value function, $Q(s, a)$:

$$Q_\pi(s, a) = \mathbb{E}_\pi \left[ \sum_{k=t}^{T} \gamma^{k-t} R(s_k, a_k) \middle| s_t = s, a_t = a \right], \tag{1}$$

where $T$ is the episode length. When the policy $\pi$ is optimal, the $Q$-function satisfies the *Bellman equation* (Bellman, 1954).

$$Q_*(s_t, a_t) = \begin{cases} R(s_t, a_t) + \gamma \max_{a \in \mathcal{A}} Q_*(s_{t+1}, a) & \text{if } t < T \\ R(s_T, a_T) & \text{if } t = T \end{cases}. \tag{2}$$

The $Q_*()$ function can be learned using Q-learning algorithm (Watkins & Dayan, 1992). The optimal policy can be computed from $Q_*$ greedily, $\pi_*(s_t) = \arg\max_{a \in \mathcal{A}} Q_*(s_t, a)$. In *Deep Q-Networks* (DQN), Mnih et al. (2013) formulate a loss function based on the Bellman equation to approximate the optimal $Q_*$-function using a deep neural network, $Q_m$:

$$\mathcal{L}(\theta_{Q_m}) = \mathbb{E}_{a_t \sim \pi(s_t; \theta_{\pi_m})} \left[ (Q_m(s_t, a_t; \theta_{Q_m}) - y_t)^2 \right], \tag{3}$$

where $y_t = R(s_t, a_t) + \max_a \gamma Q_{\text{tgt}}(s_{t+1}, a; \theta_{Q_{\text{tgt}}})$, is the *target* and $Q_{\text{tgt}}$ is the target network (Mnih et al., 2015a). The target network is a slower-changing copy of the main network that stabilizes learning. Mnih et al. (2015a) also employ *replay buffers* (Lin, 1993) that store transitions from episodes. During training, these transitions are sampled out of order to train the networks in an off-policy manner, avoiding correlation in the samples and thus leading to faster learning.

In this work, we use an extension of DQN to continuous action spaces called deep deterministic policy-gradients (DDPG) (Lillicrap et al., 2015). DDPG approximates the policy with a policy network $\pi_{\text{tgt}}(s; \theta_\pi)$ that replaces the $\max$ operator in $y_t$. The target thus becomes $y_t =$

$R(s_t, a_t) + \gamma Q_{\text{tgt}}(s_{t+1}, \pi_{\text{tgt}}(s_{t+1}; \theta_\pi); \theta_{Q_{\text{tgt}}})$ and the loss function changes accordingly:

$$\mathcal{L}(\theta_Q, \theta_\pi) = \mathbb{E}_{a_t \sim \pi(s_t; \theta_\pi)}[(Q_m(s_t, a_t; \theta_Q) - y_t)^2]. \tag{4}$$

## 3.2 MULTI-GOAL REINFORCEMENT LEARNING

Multi-Goal Reinforcement Learning (Plappert et al., 2018) focuses on problems where the desired goal state can change for every episode. State-of-the-art MGRL algorithms learn a goal-conditioned value function (GCVF), $Q(s, a, g)$, which is defined similar to the $Q$-function (5), but with an additional dependence on the desired goal specification $g \in \mathcal{G}$ :

$$Q_\pi(s, a, g) = \mathbb{E}_\pi \left[ \sum_{k=t}^{T} \gamma^{k-t} R(s_k, a_k, g) \middle| s_t = s, a_t = a \right]. \tag{5}$$

The structure of the goal specification, $g \in \mathcal{G}$, can be arbitrary. For example, in a robotic arm experiment, possible goal specifications include the desired position of the end-effector and the desired joint angles of the robot. The states and *achieved goals* are assumed to be an observable part of the Goal-MDP to enable the agent to learn the correspondences between them, $[(s_0, a_0, g_0, r_0), \ldots, (s_T, a_T, g_T, r_T)]$. Consequently, this Goal-MDP is fully governed by the six tuple $(\mathcal{S}, \mathcal{A}, \mathcal{G}, T, R, \gamma)$. The reward, $R(s, a, g) : \mathcal{S} \times \mathcal{A} \times \mathcal{G} \to \mathbb{R}$, and policy $\pi(s, g) : \mathcal{S} \times \mathcal{G} \to \mathcal{A}$ are also in turn conditioned on goal information.

**Hindsight Experience Replay** HER (Andrychowicz et al., 2017) builds upon this definition of GCVFs (5). The main insight of HER is that there is no valuable feedback from the environment when the agent does not reach the goal. This is further exacerbated when goals are sparse in the state-space. HER solves this problem by reusing these failed experiences for learning. It recomputes a reward for each reached state by relabeling them as pseudo-goals.

In our experiments, we employ HER's *future* strategy for pseudo-goal sampling. More specifically, two transitions from the same episode in the replay buffer for times $t$ and $t + f$ are sampled. The achieved goal $g_{t+f}$ is then assumed to be the pseudo-goal. The algorithm generates a new transition for the time step $t$ with the reward re-computed as if $g_{t+f}$ was the desired goal, $(s_t, a_t, s_{t+1}, R(s_t, a_t, g_{t+f}))$. HER uses this new transition as a sample.

## 4 PATH REWARD-BASED GCVFS

In our definition of the GCVF, instead of making the reward function depend upon the goal, we count accumulated rewards over a path, *path-rewards*, only if the goal is reached. This makes the dependence on the goal explicit instead of implicit to the reward formulation. Mathematically,

$$Q_\pi^P(s, a, g^*) = \begin{cases} \mathbb{E}_\pi \left[ \sum_{k=t}^{l-1} \gamma^{k-t} R^P(s_k, a_k) \middle| s, a, g_l = g^* \right] & \text{if } \exists\, l \text{ such that } g_l = g^* \quad (6a) \\ -\infty & \text{otherwise,} \quad (6b) \end{cases}$$

where $l$ is the time step when the agent reaches the goal. If the agent does not reach the goal, the GCVF is defined to be negative infinity. This first term (6a) is the expected cumulative reward over paths from a given start state to the goal. This imposes the constraint that cyclical paths in the state space must have negative cumulative reward for (6a) to yield finite values. For most practical physical problems, this constraints naturally holds if reward is taken to be some measure of negative energy expenditure. For example, in the robot arm experiment, moving the arm must expend energy (negative reward). Achieving a positive reward cycle would translate to generating infinite energy . In all our experiments with this formulation, we use a constant reward of -1 for all states, $R^P(s, a) = -1 \,\forall s, a$.

For the cases when the agent does reach the goal at time step $l$ (6a), the Bellman equation takes the following form:

$$Q_*^P(s_t, a_t, g^*) = \begin{cases} R^P(s_t, a_t) + \gamma \max_{a \in \mathcal{A}} Q_*^P(s_{t+1}, a, g^*) & \text{if } t < l - 1 \quad (7a) \\ R^P(s_{l-1}, a_{l-1}) & \text{if } t = l - 1. \quad (7b) \end{cases}$$

Notice that terminal step in this equation is the step to reach the goal. This differs from Equation (3), where the terminal step is the step at which the episode ends. This formulation is equivalent to the end of episode occuring immediately when the goal is reached. This reformulation does not require goal-rewards, which in turn obviates the requirement for pseudo-goals and reward recomputation.

**One-Step Loss**  To enable algorithms like HER to work under this reformulation we need to recognize when the goal is reached (7b). This recognition is usually done by the reception of high goal reward. Instead, we use (7b) as a *one-step loss* that serves this purpose which is one of our main contributions:

$$\mathcal{L}_{\text{step}}(\theta_Q) = (Q_*^P(s_{l-1}, a_{l-1}, g_l; \theta_Q) - R(s_{l-1}, a_{l-1}))^2. \tag{8}$$

This loss is based on the assumption that one-step reward is the highest reward between adjacent start-goal states and allows us to estimate the one-step reward between them. Once learned, it serves as a proxy for the reward to the last step to the goal (7b). The Bellman equation (7), serves as a one-step rollout to combine rewards to find maximum reward paths to the goal.

One-step loss is different from the terminal step of Q-Learning because one-step loss is applicable to every transition unlike the terminal step. However, one-step loss can be thought of as Q-Learning where every transition is a one-step episode where the achieved goal is the pseudo goal.

One-step loss also different from the terminal condition in Kaelbling (1993). Kaelbling (1993) defines $Q_*^P(.,.,.)$ similar to Eq (7) except the terminal condition is defined as $Q_*^P(s_t, a, g*) = 0$ when $g_t = g^*$. Under the stated assumptions, the two definitions are equivalent but one-step loss is advantageous as it can be applied to every transition unlike the Kaelbling (1993) terminal condition which can be only applied when $g_t = g^*$.

We modify an implementation of HER to include the step-loss term and disable goal rewards for our experiments. As in HER, we use the DDPG loss $\mathcal{L}_{\text{ddpg}}$ while using the "future" goal sampling strategy described in the paper. The details of the resulting algorithm are shown as pseudo-code in Algorithm 1 in the Appendix.

## 4.1 Deep Floyd-Warshall Reinforcement Learning

The GCVF redefinition and one step-loss introduced in this paper are inspired by the tabular formulation of Floyd-Warshall Reinforcement Learning (FWRL) (Kaelbling, 1993). We extend this algorithm for use with deep neural networks. Unfortunately, the algorithm itself does not show significant improvement over the baselines. However, the intuitions gained in its implementation led to the contributions of this paper.

The core contribution of FWRL is a compositionality constraint in the space of GCVFs. This constraint states that the optimal $Q_*$ value from any state $s_t$ to any goal $g_{t+f}$ is greater than or equal to the sum of optimal $Q_*$ values via any intermediate state-goal pair $(s_w, g_w)$:

$$Q_*(s_t, a_t, g_w) + Q_*(s_w, \pi_*(s_w, g_{t+f}; \theta_\pi), g_{t+f}) \geq Q_*(s_t, a_t, g_{t+f}). \tag{9}$$

We translate these constraints into loss terms and add them to the DDPG loss $\mathcal{L}_{\text{ddpg}}$ and one-step loss $\mathcal{L}_{\text{step}}$. Taking cue from Mnih et al. (2015b), we do not repeat the the main online network $Q_m$ in the loss term. We use a target network $Q_{\text{tgt}}$ and split the constraint into two loss terms. One loss term acts as a lower bound $\mathcal{L}_{\text{lo}}$ and the other acts as an upper bound $\mathcal{L}_{\text{up}}$:

$$\mathcal{L}_{\text{lo}} = \text{ReLU}[Q_{\text{tgt}}(s_t, a_t, g_w) + Q_{\text{tgt}}(s_w, \pi_t(s_w, g_{t+f}; \theta_\pi), g_{t+f}) - Q_m(s_t, a_t, g_{t+f})]^2 \tag{10}$$

$$\mathcal{L}_{\text{up}} = \text{ReLU}[Q_m(s_t, a_t, g_w) + Q_{\text{tgt}}(s_w, \pi_t(s_w, g_{t+f}; \theta_\pi), g_{t+f}) - Q_{\text{tgt}}(s_t, a_t, g_{t+f})]^2. \tag{11}$$

Note that the above terms differ only by choice of the target and main network.

**FWRL Sampling**  We augment HER sampling to additionally get the intermediate state-goal pair $(s_w, g_w)$. Once a transition $(s_t, a_t, r_t, s_{t+1})$ and a future goal $g_{t+f}$ have been sampled from the same episode, we sample another intermediate state and goal pair $(s_w, g_w)$ such that $t \leq w \leq t + f$.

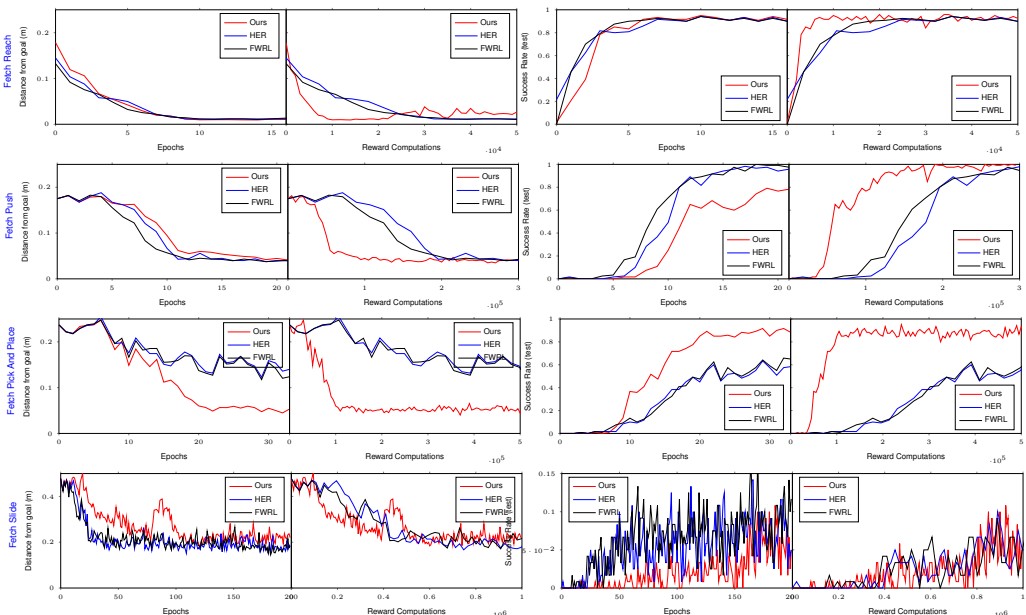

Figure 2: For the Fetch tasks, we compare our method (red) against HER (blue) (Andrychowicz et al., 2016) and FWRL (green) (Kaelbling, 1993) on the distance-from-goal and success rate metrics. Both metrics are plotted against two progress measures: the number of training epochs and the number of reward computations. Except for the Fetch Slide task, we achieve comparable or better performance across the metrics and progress measures.

# 5 EXPERIMENTS

We use the environments introduced in Plappert et al. (2018) for our experiments. Broadly the environments fall in two categories, Fetch and Hand tasks. Our results show that learning is possible across all environments without the requirement of goal-reward. More specifically, the learning happens even when reward given to our algorithm is agent is always "-1" as opposed to the HER formulation where a special goal-reward of "0" is needed for learning to happen.

The Fetch tasks involve a simulation of the Fetch robot's 7-DOF robotic arm. The four tasks are Reach, Push, Slide and PickAndPlace. In the Reach task the arm's end-effector is tasked to reach the a particular 3D coordinate. In the Push task a block on a table needs to be pushed to a given point on it. In the Slide task a puck must be slid to a desired location. In the PickAndPlace task a block on a table must be picked up and moved to a 3D coordinate.

The Hand tasks use a simulation of the Shadow's Dexterous Hand to manipulate objects of different shapes and sizes. These tasks are HandReach, HandManipulateBlockRotateXYZ, HandManipulateEggFull and HandManipulatePenRotate. In HandReach the hand's fingertips need to reach a given configuration. In the HandManipulateBlockRotateXYZ, the hand needs to rotate a cubic block to a desired orientation. In HandManipulateEggFull, the hand repeats this orientation task with an egg, and in HandManipulatePenRotate, it does so with a pen.

Snapshots of all these tasks can be found in Figure 1. Note that these tasks use joint angles, not visual input.

## 5.1 METRICS

Similar to prior work, we evaluate all experiments on two metrics: the success rate and the average distance to the goal. The success rate is defined as the fraction of episodes in which the agent is able to reach the goal within a pre-defined threshold region. The metric *distance of the goal* is the euclidean distance between the achieved goal and the desired goal in meters. These metrics are

plotted against a standard progress measure, the number of training epochs, showing comparable results of our method to the baselines.

To emphasize that our method does not require goal-reward and reward re-computation, we plot these metrics against another progress measure, the number of reward computations used during training. This includes both the episode rollouts and the reward recomputations during HER sampling.

## 5.2 HYPER-PARAMETERS CHOICES

Unless specified, all our hyper-parameters are identical to the ones used in the HER implementation (Dhariwal et al., 2017). We note two main changes to HER to make the comparison more fair. Firstly, we use a smaller *distance-threshold*. The environment used for HER and FWRL returns the goal-reward when the achieved goal is within this threshold of the desired goal. Because of the absence of goal-rewards, the distance-threshold information is not used by our method. We reduce the threshold to 1cm which is reduction by a factor of 5 compared to HER.

Secondly, we run all experiments on 6 cores each, while HER uses 19. The batch size used is a function of the number of cores and hence this parameter has a significant effect on learning.

To ensure fair comparison, all experiments are run with the same hyper-parameters and random seeds to ensure that variations in performance are purely due to differences between the algorithms.

## 5.3 RESULTS

All our experimental results are described below, highlighting the strengths and weaknesses of our algorithm. Across all our experiments, the distance-to-the-goal metric achieves comparable performance to HER *without requiring goal-rewards*.

**Fetch Tasks** The experimental results for Fetch tasks are shown in Figure 2. For the Fetch Reach and Push tasks, our method achieves comparable performance to the baselines across both metrics in terms of training epochs and outperforms them in terms of reward recomputations. Notably, the Fetch Pick and Place task trains in significantly fewer epochs. For the Fetch Slide task the opposite is true. We conjecture that Fetch Slide is more sensitive to the distance threshold information, which our method is unable to use.

**Hand Tasks** For the Hand tasks, the distance to the goal and the success rate show different trends. We show the results in Figure 3. When the distance metric is plotted against epochs, we get comparable performance for all tasks; when plotted against reward computations, we outperform all baselines on all tasks except Hand Reach. The baselines perform well enough on this task, leaving less scope for significant improvement. These trends do not hold for the success rate metric, on which our method consistently under-performs compared to the baselines across tasks. This is surprising, as all algorithms average equally on the distance-from-goal metric. We conjecture that this might be the result of high-distance failure cases of the baselines, i.e. when the baselines fail, they do so at larger distances from the goal. In contrast, we assume our method's success and failure cases are closer together.

## 6 ANALYSIS

To gain a deeper understanding of the method we perform three additional experiments on different tasks. We ask the following questions: (a) How important is the step loss? (b) What happens when the goal-reward is also available to our method? (c) How sensitive is HER and our method to the distance-threshold?

**How important is the step loss?** We choose the Fetch-Push task for this experiment. We run our algorithm with no goal reward and without the step loss on this task. Results show that our algorithm fails to reach the goal when the step-loss is removed (Fig. 4a) showing its necessity.

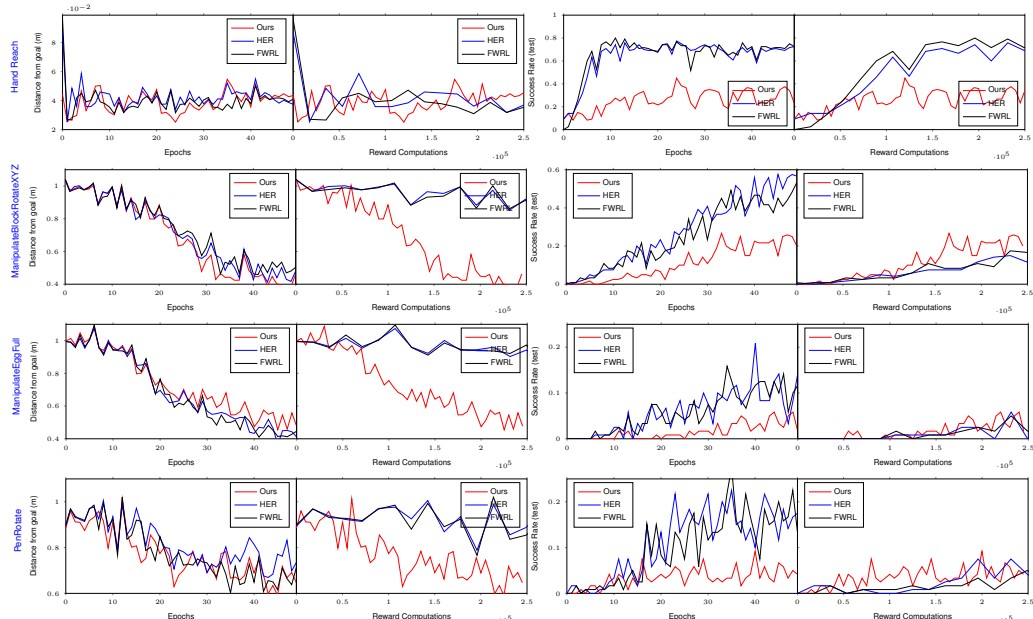

Figure 3: For the hand tasks, we compare our method (red) against HER (blue) (Andrychowicz et al., 2016) and FWRL (green) (Kaelbling, 1993) for the distance-from-goal and success rate metrics. Furthermore, both metrics are plotted against two progress measures, the number of training epochs and the number of reward computations. Measured by distance from the goal, our method performs comparable to or better than the baselines for both progress measurements. For the success rate, our method underperforms against the baselines.

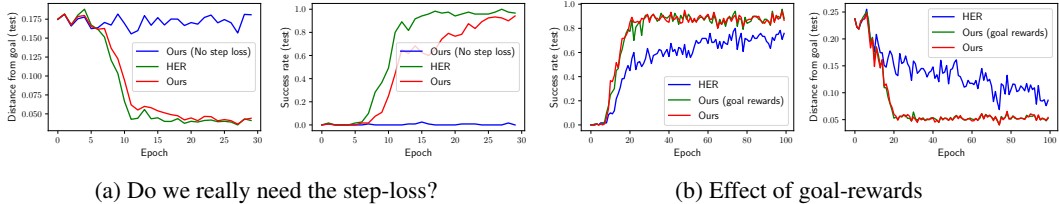

(a) Do we really need the step-loss?

(b) Effect of goal-rewards

Figure 4: (a) Effects of removing the step-loss from our methods. Results show that it is a critical component to learning in the absence of goal-rewards. (b) Adding goal-rewards to our algorithm that does have an effect further displaying how they are avoidable.

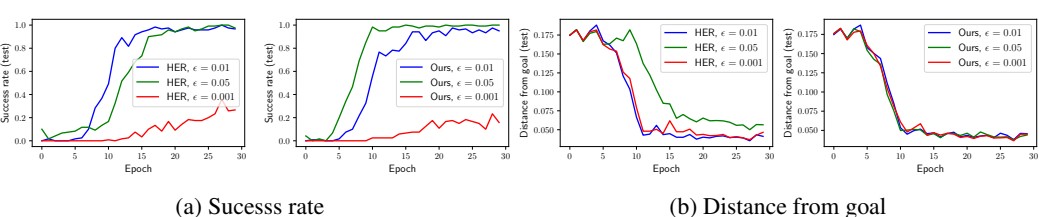

(a) Sucesss rate

(b) Distance from goal

Figure 5: We measure the sensitive of HER and our method to the dsitance-threshold ($\epsilon$) with respect to the success-rate and distance-from-goal metrics. Both algorithms success-rate is sensitive the threshold while only HER's distance-from-goal is affected by it.

**What happens when the goal-reward is also available to our method?** We run this experiment on the Fetch PickAndPlace task. We find that goal-rewards do not affect the performance of our algorithm further solidifying the avoidability of goal-reward (Fig 4b).

**How sensitive is HER and our method to the distance-threshold?** In the absence of goal-rewards, our algorithm is not to able capture distance threshold information that decides whether the agent has reached the goal or not. This information is available to HER. To understand the sensitivity of our algorithm and HER on this parameter, we vary it over 0.05 (the original HER value), 0.01 and 0.001 meters (Fig. 5). Results show that for the success-rate metric, which is itself a function of this parameter, both algorithms are affected equally (Fig. 5a). For the distance-from-goal, only HER is affected (Fig. 5b). This fits our expectations as set up in section 5.2.

## 7 CONCLUSION

In this work we pose a reinterpretation of goal-conditioned value functions and show that under this paradigm learning is possible in the absence of goal reward. This is a surprising result that runs counter to intuitions that underlie most reinforcement learning algorithms. In future work, we will augment our method to incorporate the distance-threshold information to make the task easier to learn when the threshold is high. We hope that the experiments and results presented in this paper lead to a broader discussion about the assumptions actually required for learning multi-goal tasks.

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

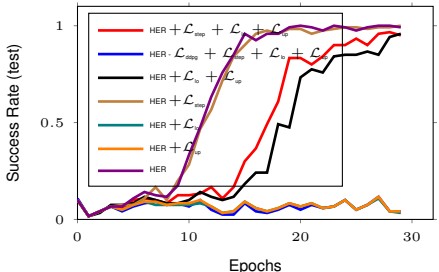 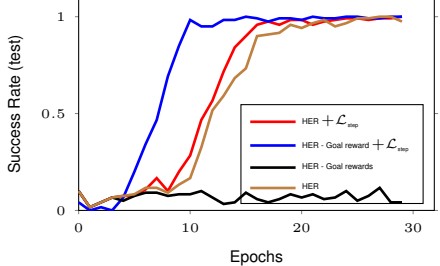

Figure 6: Ablation on loss functions for Fetch Push task. The Floyd-Warshall inspired loss functions $\mathcal{L}_{\text{lo}}$ and $\mathcal{L}_{\text{up}}$ do not help much. $\mathcal{L}_{\text{step}}$ helps a little but only in conjunction with HER Andrychowicz et al. (2016).

Figure 7: Even when the Goal rewards are removed from HER Andrychowicz et al. (2016) training, the HER is able to learn only if the $\mathcal{L}_{\text{step}}$ is added again. (HER-Goal Rewards+$\mathcal{L}_{\text{step}}$) is our proposed method.

## 8 APPENDIX

Our algorithm 1 is different from HER Andrychowicz et al. (2016) because it contains additional step-loss term $\mathcal{L}_{\text{step}}$ at line number 17 which allows the algorithm to learn even when the rewards received are independent of desired goal. Also in HER sampling (line 13), the algorithm recomputes the rewards because the goal is replaced with a pseudo-goal. Our algorithm does not need reward recomputation because the reward formulation does not depend on the goal and is not affected by choice of pseudo-goal. Our algorithm is also different from Floyd-Warshall Reinforcement learning because it does not contain $\mathcal{L}_{\text{up}}$ and $\mathcal{L}_{\text{lo}}$ terms and contains the additional $\mathcal{L}_{\text{step}}$.

---

**Algorithm 1:** Path-reward reinforcement learning

```
   /* By default all states are unreachable                                      */
 1 Initialize networks Q_m (s_i, a_i, g_j; θ_Q) and π(s_i, s_g; θ_π) ;
 2 Copy the main network to target network Q_tgt (s_i, a_i, g_j; θ_Q) ← Q_m (s_i, a_i, g_j; θ_Q) ;
 3 Initialize replay memory M ;
 4 for e ← 1 to E do
 5       Sample g_e ∈ 𝒢 ;
 6       Set t ← 0;
 7       Observe state s_t and achieved goal g_t ;
         /* Episode rollout                                                       */
 8       for t ← 1 to T do
 9            Take action a_t ← ε-greedy(π_m(s_t, g; θ_π)) ;
10            Observe s_{t+1}, g_{t+1}, r_t ;
11            Store (s_t, g_t, a_t, s_{t+1}, g_{t+1}, r_t; g_e) in memory M[e] ;
         /* Train                                                                 */
12       for t ← 1 to T do
13            HER sample batch B = [(s_i, g_i, a_i, s_{i+1}, g_{i+1}, r_i; g_{i+f_i}), ..., (s_b, g_b, a_b, s_{b+1}, g_{b+1}, r_b; g_{b+f_b})] from M ;
14            𝓛(...) = 0 ;
15            for b ∈ 1 to |B| do
16                 (s_b, g_b, a_b, s_{b+1}, g_{b+1}, r_b, g_{b+f_b}) = B[b] ;
                   /* Step loss                                                   */
17                 𝓛(...) += (Q_m (s_b, a_b, g_{b+1}) − r_b)² ;
                   /* DDPG loss                                                   */
18                 𝓛(...) += (Q_m (s_b, a_b, g_{b+f_b}) − r_b − γQ_tgt (s_{b+1}, π_tgt(s_{b+1}, g_{b+f_b}; θ_π), g_{b+f_b}))² ;
19            Update gradients for Q_m and π_m using loss 𝓛(...);
   Result: Q_m, π_m
```

---

## 9 ABLATION ON LOSS AND GOAL REWARDS

In Figure 6 and Figure 7 we show ablation on loss functions and goal rewards. In Figure 7 Our method is shown in blue with HER - Goal rewards + $\mathcal{L}_{\text{step}}$.

