# OpenReview forum: "Learning Goal-Conditioned Value Functions with one-step Path rewards rather than Goal-Rewards"
_ICLR.cc/2019/Conference_

### Official Review · AnonReviewer2 · 2018-11-02
**Evaluation and judgement**

**Rating:** 3
**Confidence:** 4

**Review:**

The paper presents an approach for an approach to addressing multi-goal reinforcement learning, based on what they call "one-step path rewards" as an alternative to the use of goal conditioned value function.
The idea builds on an extension of a prior work on FWRL.
The paper presents empirical comparison of the proposed method with two baselines, FWRL and HER.
The experimental results are mixed, and do not convincingly demonstrate the effectiveness/superiority of the proposed method.
The idea of the proposed method is relatively simple, and is not theoretically justified.

Based on these observations, the paper falls short of the conference standard.

---

> ### Author Response · Authors · 2018-11-08
> **Our method consistent performs better than baselines when computered in terms of distance from the goal and reward computation**
>
> Thank you for your feedback.
>
> 1. The experimental results are mixed, and do not convincingly demonstrate the
>    effectiveness/superiority of the proposed method.
>
> The results are mixed when the learning curves are compared with respect to the epochs (the number of transition samples) that intentionally does not take reward-recomputation in to account.
>
> When this computation is taken in to account, our algorithm comprehensively improves upon the baselines in 6 out of 8 experiments. To further highlight these differences we magnify our reward-recomputation plots to eliminate sections where curves overlap and are non-informative. These changes can be found in Figure 2 and 3.
>
> We further reiterate that reward recomputation
> cost can be significant dependent upon the environment and setup. In cases when
> the reward-computation depends upon collisions and haptic feedback of real
> robots, the reward recomputation may even be impossible without re-running the
> experiment. Hence reducing reward-computation based on a simple loss term is an
> important contribution.
>
>
> 2. The idea of the proposed method is relatively simple, and is not theoretically justified.
>
> The main contribution of this paper is to show that goal-conditioned value functions can be learned without requiring goal-reward.
> We believe that the simplicity of this proposed idea is the beauty of the method leading to significant changes in performance of the algorithm when reward recomputation is taken in to account.
>
> Our algorithm builds upon HER which does not itself possess theoretical guarantees. We would be able to addess this point specifically if the reviewer could clarify what kind of theoretical justification they would expect to see.

---

### Official Review · AnonReviewer3 · 2018-11-02

**Rating:** 1
**Confidence:** 4

**Review:**

This paper aims to improve on Hindsight Experience Replay by removing the need to compute rewards for reaching a goal. The idea is to frame goal-reaching as a shortest path problem where all rewards are -1 until the goal is reached, removing the need to compute rewards. While similar ideas were explored in a recent arxiv tech report, this paper claims to build on these ideas with new loss functions. The experimental results do not seem to be any better compared to baselines when measured in terms of data efficiency, but the proposed method requires fewer “reward computations”.

Clarity:
While the ideas in the paper were easy to follow, there are a number of problems with the writing. The biggest problem is that it wasn’t clear exactly what algorithms were evaluated in the experiments. There is an algorithm box for the proposed method in the appendix, but it’s not clear how the method differs from the FWRL baseline.

Another major problem is that the paper does a poor job of citing earlier related work on RL. DQN is introduced without mentioning or citing Q-learning. Experience replay is mentioned without citing the work of Long-Ji Lin. There’s no mention of earlier work on shortest-path RL from LP Kaelbling from 1993.

Novelty and Significance:
After reading the paper I am not convinced that there’s anything substantially new in this paper. Here are my main concerns:

1) The shortest path perspective for goal-reaching was introduced in “Learning to Achieve Goals” by LP Kaelbling [1]. This paper should be cited and discussed.

2) I am not convinced that the proposed formulation is any different than what is in Hindsight Experience Replay (HER) paper. Section 3.2 of the HER paper defines the reward function as -1 if the current state is not the same as the goal and 0 if the current state is the same as the goal. Isn’t this exactly the cost-to-go/shortest path reward structure that is used in this paper?

3) This paper claims that the one-step loss (Equation 8) is new, but it is actually the definition of the Q-learning update for transitioning to a terminal state. Since goal states are absorbing/terminal, any transition to a goal state must use the reward as the target without bootstrapping. So the one-step loss is just Q-learning and is not new. This is exactly how it is described in Section 3 of [1].

4) The argument that the proposed method requires fewer reward evaluations than FWRL or HER seems flawed. HER defines the reward to be -1 if the current state and the goal are different and 0 if they are the same. As far as I can tell this paper uses the same reward structure, so how is it saving any computation?

Can the authors comment on these points and clarify what they see as the novelty of this work?

Overall quality:
Unless the authors can convince me that the method is not equivalent to existing work I don’t see enough novelty or significance for an ICLR paper.

[1] “Learning to Achieve Goals” LP Kaelbling, 1993.

---

> ### Author Response · Authors · 2018-11-08
> **In our reward formulation we do not get 0 reward, it is always -1**
>
> There are two main reasons for the confusion about the contributions of
> this work.
>
> First, our reward formulation is different from that of Hindsight Experience Replay
> (HER). In HER, the agent receives -1 reward for all state transitions except on
> reaching the goal when it receives 0 reward. In contrast, for our reward
> formulation the agent receives -1 reward for all state transitions including
> when agent reaches and continues to stay at the goal.
>
> Second, our reward formulation is atypical with respect to conventional
> Reinforcement Learning (RL). In conventional RL, a high reward is used to
> specify the desired goal (goal-reward). However, this goal-reward is not
> necessary in goal-conditioned RL because the goal specification is already given
> at the start of every episode. We believe that this result is counter-intuitive
> and will be interesting to the RL community.
>
> We clarify the reviewer's concerns and
> edit our draft to minimize chances of similar confusion.
>
> Clarity:
> 1. The main difference between HER, FWRL and our algorithm lies in the choice of
>    loss terms used. HER uses Eq (3), FWRL uses Eq (3) + L_up + L_lo, and Our
>    algorithm uses Eq (3) + L_step as shown in the pseudo-code. Another difference
>    is due to reward formulation. Because our rewards are independent of reaching
>    the goal, we do not need to recompute rewards. We have added the description
>    about these differences in the appendix to highlight them.
>
> 2. We have introduced the requested citations at appropriate places in the
>    paper.
>
>    Since [1] introduced the idea of FWRL before Dhiman et. al. 2018,
>    we replace the attributions accordingly in the paper. We further add
>    discussion points specific to their algorithm in the Related Work and
>    One-Step Loss section.
>
> Novelty and Significance
> 1. Our contribution is learning *without* using goal-rewards *using* the
>    shortest path perspective. Our secondary contribution is to extend [1] to
>    deep neural networks.
>
> 2. As mentioned above, our reward is always -1 *even when* current state is same
>    as the goal state.
>
>    Similar to HER, our goal states are not absorbing/terminal. Instead the
>    episodes are of fixed number of steps and the agent is encouraged to stay in
>    the goal state for as long as possible. This is how the replay buffer is
>    populated and how the average episode reward is computed. However, the
>    objective maximized is equivalent to treating this fixed length episode
>    problem as if the episodes are terminating on reaching the goal.
>
>    To further clarify this in the paper, we have added reward structure
>    details to the Introduction (section 1, paragraph 3) and the
>    Experiments section (section 5, end of paragraph 1).
>
> 3. One-step loss is different from both the terminal step of both Q-Learning and [1].
>
>    One-step loss is different from terminal step of Q-Learning because it is
>    applied to every state transition rather than just the terminal step of the
>    episode. Having said that it is indeed equivalent to Q-Learning, if every
>    state transition is viewed as a one-step episode with the reached state as
>    the pseudo-goal. Correspondingly we have updated the manuscript in both the
>    introduction and the one-step loss section to include one-step-episode
>    Q-Learning perspective.
>
>    One-step loss is also different from the terminal step of [1].
>    Referring to Section 3 of [1], we see the one-step loss (Eq. 8)
>    as an alternative of the terminal condition DG*(s, a, g) = 0 if s = g in the
>    recursive definition of DG*(s, a, g). As an alternative, one-step loss
>    translates to DG*(s_t, a_t, g_{t+1}) = -1, for all transitions (s_t, a_t ->
>    g_{t+1}) i.e. it removes the dependence of checking s=g. Although it serves
>    the same purpose of terminal condition in recursive definition but the
>    condition is mathematically different and requires the different
>    assumption that one-step path is the highest reward path between s_t and g_{t+1}.
>
>
> 4. As stated earlier, our reward independent of desired goal. The reward
>    re-computation for the pseudo-goals becomes unnecessary because the reward
>    does depend upon the check if current state is same as the desired goal.
>
>    To further highlight saved reward computation, we magnify on our
>    reward-computation plots removing the uninformative parts of the plots where
>    the curves overlap.
>
> Overall quality:
>
> (A) Novelty: As argued above our proposed one-step loss is novel and so is the
> extension of [1] from tabular domain to deep learning.
>
> (B) Significance
>   (a) The counter-intuitive result that goal-conditioned RL does not need goal
>   reward is worth bringing to the attention of the RL community
>   (b) The absence of the requirement of reward-recomputation is significant
>   because in real robotics experiments, the reward computation may not be
>   possible without re-running the entire experiment.
>
> [1]: Kaelbling, Leslie Pack. "Learning to achieve goals." IJCAI. 1993.

---

> > ### Comment · AnonReviewer3 · 2018-11-23
> > **response**
> >
> > Thank you for the clarifications. I think they confirmed my understanding of the paper.
> >
> > I maintain that the idea that you can do goal-based learning without rewards is both in the “Hindsight Experience Replay” paper and in the “Learning to Achieve Goals” paper.
> >
> > Here’s what the HER paper says about a trajectory s_1, …, s_T for a goal g (top of page 4):
> > “The pivotal idea behind our approach is to re-examine this trajectory with a different goal — while this trajectory may not help us learn how to achieve the state g, it definitely tells us something about how to achieve the state s_T . This information can be harvested by using an off-policy RL algorithm and experience replay where we replace g in the replay buffer by s_T”.
> > I think it is clear that by replacing the goal with the final (reached) state s_T one can just give a reward of 0 at the final transition and -1 to the preceding ones. There is no need to compute rewards or compare any states.
> >
> > This is exactly what your one-step loss does. It is equivalent to a Q-learning update on each transition s,a,s’ with the goal relabeled to s’. The transition becomes terminal since the goal is reached. Presenting the one-step loss as something new is not accurate.
> >
> > Having said that, there are multiple ways of instantiating this idea. The HER paper chooses to relabel goals for a trajectories. So in a sense the one-step loss is applied only to the last transition. You propose to apply the relabeling to all transitions. “Learning to Achieve Goals” performs all goal updating so it will also relabel each transition with the achieved state s’ as the goal.
> >
> > Comparing these approaches in terms of performance could be interesting, but as your results suggest there is not really a difference between HER and your approach in terms of data efficiency. I don’t buy the comparison in terms of “reward computations” because HER can also be implemented in a way where rewards don’t need to be recomputed.

---

> > > ### Author Response · Authors · 2018-11-26
> > > **There are multiple ways of instantiating an idea and ours is one**
> > >
> > > We agree that your proposed modification to HER would negate the reward re-computation requirement. However, HER does not do so. This perspective was influenced by the observation that reward re-computations are redundant. This observation must be non-trivial because HER and its many extensions have not accounted for it.
> > >
> > > > I think it is clear that by replacing the goal with the final (reached) state s_T one can just give a reward of 0 at the final transition and -1 to the preceding ones. There is no need to compute rewards or compare any states.
> > >
> > > While your solution would work, we think our solution is much simpler to implement because it does not require checking whether sampled t == T. The replay buffer is sampled as is, with only a loss term added to the loss function. These are different ways to instantiate the same ideas which we believe to be non-trivial.
> > >
> > > We thank you for your detailed comments and feedback.

---

> > > > ### Comment · AnonReviewer3 · 2018-12-04
> > > > **clarification**
> > > >
> > > > > We agree that your proposed modification to HER would negate the reward re-computation requirement. However, HER does not do so. This perspective was influenced by the observation that reward re-computations are redundant. This observation must be non-trivial because HER and its many extensions have not accounted for it.
> > > >
> > > > I am not proposing any modifications of HER. I am simply pointing out that the idea that you can do goal-based learning without recomputing rewards is both in the “Hindsight Experience Replay” paper and in the “Learning to Achieve Goals” paper. To me it is the key idea behind HER. If you've reached a state s then you've achieved the goal of reaching state s.
> > > >
> > > > > While your solution would work, we think our solution is much simpler to implement because it does not require checking whether sampled t == T. The replay buffer is sampled as is, with only a loss term added to the loss function. These are different ways to instantiate the same ideas which we believe to be non-trivial.
> > > >
> > > > I am not proposing any alternative solutions to the problem. Again, to me the key idea behind HER is that if you've reached state s, then you've achieved the goal of reaching state s. This means that the agent can get a reward of of 1 in the 0/1 reward formulation or a reward of 0 in the -1/0 formulation and the state is considered terminal. There is no need to check for equality of states and time indices.
> > > >
> > > > I maintain that they key idea behind this paper is not new. On top of that, the way it is presented obfuscates what is really going on. What is the justification for adding the 1-step loss to Q-learning with a constant reward for all transitions? Why will this converge at all? HER just applies Q-learning with modified goals/reward, and Q-learning comes with theoretical guarantees.

---

> > > > > ### Author Response · Authors · 2018-12-18
> > > > > **Response**
> > > > >
> > > > > We address your comments point by point.
> > > > >
> > > > > > I maintain that they key idea behind this paper is not new. On top of that, the way it is presented obfuscates what is really going on. What is the justification for adding the 1-step loss to Q-learning with a constant reward for all transitions? Why will this converge at all? HER just applies Q-learning with modified goals/reward, and Q-learning comes with theoretical guarantees.
> > > > >
> > > > > The surprising result that a constant reward for all transition converges at all is the main message behind our work. The reason why it converges is because we estimate the path-reward (as done by Kaelbling 1993) instead of future-reward (as done by Q-learning/HER). Our work builds upon Kalebling's work which does not have theoretical guarantees yet, but I do not see any reason why the formulation is averse to theoretical guarantees.
> > > > >
> > > > > >  I am not proposing any alternative solutions to the problem. Again, to me the key idea behind HER is that if you've reached state s, then you've achieved the goal of reaching state s. This means that the agent can get a reward of of 1 in the 0/1 reward formulation or a reward of 0 in the -1/0 formulation and the state is considered terminal. There is no need to check for equality of states and time indices.
> > > > >
> > > > > What you are describing is the "final" strategy described in HER paper Section 4.5 which we performs worse than "future" strategy. We use "future" strategy in all our experiments. In "future" strategy you have to either compare against the time-index or the goal itself. Moreover, 0 goal reward is different from no-goal reward which is what we propose.
> > > > >
> > > > > >  I am not proposing any modifications of HER. I am simply pointing out that the idea that you can do goal-based learning without recomputing rewards is both in the “Hindsight Experience Replay” paper and in the “Learning to Achieve Goals” paper. To me it is the key idea behind HER. If you've reached a state s then you've achieved the goal of reaching state s.
> > > > >
> > > > > The idea is there in "Learning to Achieve Goals" paper but not in "Hindsight Experience Replay" paper. The idea is not whether you have achieved the goal of reaching state s, but the idea is whether you should get a high-goal-reward on reaching the state s. We maintain that R(s, a, g) = 0 if s == g else -1 is unnecessary and R(s, a) = -1 is enough because only the path-rewards to reach the goal matter, not the eventual "0" reward that you get on reaching the goal.
> > > > >
> > > > > Since our experiments establish that triangular inequality from "Learning to Achieve Goals" is not helpful but the one-step loss is helpful, we bring the useful ideas from "Learning to Achieve Goals" to forefront in deep learning context. This is another way to look at our contributions.
> > > > >
> > > > > We again thank you for your detailed comments and discussion.

---

### Official Review · AnonReviewer1 · 2018-11-08

**Rating:** 4
**Confidence:** 3

**Review:**

This paper presents a reinterpretation of hindsight experience replay (HER) that avoids recomputing the reward function on resampled hindsight goals in favor of simply forcing the terminal state flag for goal-achieving transitions, referred to by the authors as a "step loss".
The new proposal is evaluated on two goal-conditioned tasks from low-dimensional observations, and show modest improvements over HER and a function-approximation version of Floyd-Warshall RL, mostly as measured against the number of times the reward function needs to be recomputed.

Pros:
- minor improvement in computational cost
- investigation of classical FWRL technique in context of deep RL

Cons:
- computational improvement seems very minor
- sparse-reward implementations of HER already do essentially what this paper proposes

Comments:

The main contribution of the paper appears to be the addition of what the authors refer to as a "step loss", which in this case enforces the Q function to correctly incorporate the termination condition when goals are achieved. I.E. the discounted sum of future rewards for states that achieve termination should be exactly equal to the reward at that timestep.

It's not clear to me how this is fundamentally different than HER. One possible "sparse reward" implementation of HER involves no reward function recomputation at all, instead simply replacing the scalar reward and termination flag for resampled transitions with the indicator function for whether the transition achieves the resampled goal.
Is this not essentially identical to the proposal in this paper? I would consider this a task-dependent implementation detail for an application of HER rather than a research contribution that deserves an entire paper.

The authors claim the main advantage here is avoiding recomputation of the reward function for resampled goals.
I do not find this particularly compelling, given that all of the evaluations are done in low-dimensional state space: reward recomputation here is just a low-dimensional euclidean distance computation followed by a simple threshold.
In a world where we're doing millions of forward and backward passes of large matrix multiplications, is this a savings that really requires investigation?
It is somewhat telling that the results are compared primarily in terms of "# of reward function evaluations" rather than wall time. If the savings were significant, I expect a wall time comparison would be more compelling.
Maybe the authors can come up with a situation in which reward recomputation is truly expensive and worth avoiding?

All of the experiments in this paper use a somewhat unusual task setup where every timestep has a reward of -1. Have the authors considered other reward structures, such as the indicator function R=(1 if s==g else 0) or a distance-based dense reward?
Would this proposal work in these cases? If not, how significant is a small change to HER if it can only work for one specific reward function?

Conclusion:

In my view, the main contribution is incremental at best, and potentially identical to many existing implementations of HER.
The reconsideration of Floyd-Warshall RL in the context of deep neural networks is a refreshing idea and seems worth investigating, but I would need to see much more careful analysis before I could recommend this for publication.

---

> ### Author Response · Authors · 2018-11-20
> **One-step loss is applicable to all transitions not just terminal condition**
>
> > The main contribution of the paper appears to be ...  equal to the reward at that timestep.
>
> The one-step loss is, in fact, incorporated for every transition between states, not just the termination condition when the goal is achieved. An alternative perspective of one-step loss is one-step-episode Q-Learning. In other words, the one-step loss function is equivalent to treating every state transition as a full episode and the terminating condition. In the paper we have updated the "one-step loss" section to include this perspective.
>
> > It's not clear to me how this is fundamentally different than HER ... the transition achieves the resampled goal.
>
> All our comparisons are already with "sparse reward" R(s,a,g) = (0 if s == g else -1) implementation of HER. As far as we can understand, in your proposed formulation the reward should be R(s,a,g) = (1 if s == g else 0) which is shifted by a constant factor. The sparse reward formulation still possesses the unnecessary dependence on the goal whose redundancy and removal is the emphasis of our work.
>
> > Is this not essentially identical to the proposal in this paper? ... deserves an entire paper.
>
> No, this is not identical to the paper. At no point in our algorithm do we check the condition s == g. The proposed one-step loss that learns one-step reward Q(s_t, a_t, g=s_{t+1}) = r_t as we apply one-step loss to every transition. One-step loss is therefore task independent. As mentioned previously, this can also be thought of as one-step hindsight experience replay where the achieved goal at every step is treated as the desired goal.
>
> > The authors claim the main advantage here is avoiding recomputation of the reward function for resampled goals ... worth avoiding?
>
> In machine learning, the sample complexity is always distinguished from computation complexity. The only case where the two are comparable is when the samples are generated from simulations which is, admittedly, true for our experiments. However, our proposed improvement is general enough to be applicable to non-simulation experiments.
>
> It is a consequence of this task-dependent reward formulation that it can be re-sampled cheaply, hence, the computational cost is improvement is marginal. But we eliminate a redundancy common to the HER algorithm and its derivatives. With the massive popularity of HER (107 citations and counting), we believe that this is a worthwhile contribution to bring to the attention of the RL community.
>
> Consider the example of an agent navigating a maze where the goal is specified in the form of an image. The semantic comparison of the observed image with the goal image is an expensive operation that will require separate training for goal-dependent reward formulation [1]. However, in our proposed formulation, the comparison operation (s == g) in the reward formulation is not needed thereby eliminating the need of another learning module.
>
>
> > All of the experiments in this paper use a somewhat unusual task setup where every timestep has a reward of -1.
>
> This unusual reward formulation is possible because of our contribution (one-step loss). Hence, it is only true for the experiments that are referred to with "Ours" label. All the baselines (HER and FWRL) and "Ours (goal rewards)" operate on the reward structure for HER which is R=(0 if s==g else -1).
>
> > Have the authors considered other reward structures, such as the indicator function R=(1 if s==g else 0) or a distance-based dense reward?
> > Would this proposal work in these cases? If not, how significant is a small change to HER if it can only work for one specific reward function?
>
> We have considered and we are advocating against such reward structures because of their goal dependence. In fact in one experiment we run our algorithm with the reward structure R=(0 if s==g else -1) which is equivalent to yours with a constant shift. These results can be found in Fig. 4(b), labeled as "Ours (goal rewards)".
>
> Distance-based dense reward is by definition goal dependent. Our contribution is to eliminate this dependence. RL on dense rewards is easier than sparse rewards. Hence, we do not believe that distance-based reward adds much to our contribution. We do note that our method does work with goal based sparse rewards(Fig. 4b) and hence we would expect to continue to work with dense rewards.
>
> > The reconsideration of Floyd-Warshall RL ... recommend this for publication.
>
> We analyze FWRL and added the ablation study of loss function in Appendix Figure 6.
> It is clear that FWRL inspired loss function do not contribute to better
> learning. Instead, they hurt the performance. We think this is because Bellman inspired loss already captures the information that FWRL inspired constraints intend to capture.
>
> [1] Nikolay Savinov, Alexey Dosovitskiy, Vladlen Koltun. "Semi-Parametric Topological Memory for Navigation". In ICLR 2018

---

### Public Comment · (anonymous) · 2018-10-03
**Floyd-Warshall & RL**

The paper cites a recent arXiv paper for the concept of employing the Floyd-Warshall algorithm in goal-based reinforcement learning. This was actually introduced into the reinforcement learning literature 25 years ago in "Learning to Achieve Goals" https://people.csail.mit.edu/lpk/papers/ijcai93.ps
 by Leslie Pack Kaelbling, in IJCAI 93. However, the extension to the non-tabular case presented here does sound interesting.

---

> ### Author Response · Authors · 2018-10-03
> **We will update the attribution**
>
> Thank you for your comment. We were made aware of this paper recently. We will replace the attribution for the tabular version of the path-rewards idea with Kaelbling (1993) in an updated version of the manuscript.

---

### Meta-Review · Area_Chair1 · 2018-12-13
**Important subject matter but novelty & results insufficient for acceptance.**

**Confidence:** 4
**Recommendation:** Reject

**Metareview:**

This manuscript presents a reinterpretation of hindsight experience replay which aims to avoid recomputing the reward function, and investigates Floyd-Warshall RL in the function approximation setting.

The paper was judged as relatively clear. The authors report a slight improvement in computational cost, which some reviewers called into question. However, all of the reviewers pointed out that the experimental evidence for the method's superiority is weak. Two reviewers additionally raised that this wasn't significantly different than the standard formulation of Hindsight Experience Replay, which doesn't require the computation of rewards for relabeled goals.

Ultimately, reviewers were in agreement that the novelty of the method and quality of the obtained results rendered the work insufficient for publication. The Area Chair concurs, and urges the authors to consider the reviewers' pointers to the existing literature in order to clarify their contribution for subsequent submission.